# Simulation of Future Electric Vehicle Charging Behavior—Effects of Transition from PHEV to FEV

**Igna Vermeulen [2], Jurjen Rienk Helmus [1,*] , Mike Lees [2] and Robert van den Hoed [1]**

[1]   Urban Technology research group, Faculty of Technology, University of Applied Science Amsterdam, Weesperzijde 190 1097 DZ Amsterdam, The Netherlands; R.van.den.Hoed@hva.nl

[2]   Computational Science Lab, University of Amsterdam, Science Park 904 1098XH Amsterdam, The Netherlands; Ignavermeulen@hotmail.com (I.V.); M.H.Lees@uva.nl (M.L.)

*    Correspondence: J.R.Helmus@hva.nl; Tel.: +31-6-2115-6014

**Abstract:** The Netherlands is a frontrunner in the field of public charging infrastructure, having one of the highest number of public charging stations per electric vehicle (EV) in the world. During the early years of adoption (2012–2015), a large percentage of the EV fleet were plugin hybrid electric vehicles (PHEV) due to the subsidy scheme at that time. With an increasing number of full electric vehicles (FEVs) on the market and a current subsidy scheme for FEVs only, a transition of the EV fleet from PHEV to FEV is expected. This is hypothesized to have an effect on the charging behavior of the complete fleet, and is reason to understand better how PHEVs and FEVs differ in charging behavior and how this impacts charging infrastructure usage. In this paper, the effects of the transition of PHEV to FEV is simulated by extending an existing agent-based model. Results show important effects of this transition on charging infrastructure performance.

**Keywords:** modelling and simulation; energy supply and infrastructure; public policy and promotion; electric vehicles

---

## 1. Introduction

The Netherlands is known to be one of the frontrunners in electric vehicle (EV) adoption and public charging infrastructure rollout [1,2]. During early years of EV adoption (2012–2015), both plugin hybrid electric vehicles (PHEVs) and full electric vehicles (FEVs) were subsidized, leading to large uptake of EVs by mostly lease drivers [1–4]. Due to the limited supply of EV models and almost equal tax advantage at that time, most uptake was due to PHEVs and only limited uptake was due to FEVs (such as Tesla and Nissan Leaf) [5–7]. In the year 2018 it is expected that many car manufacturers will launch FEVs on the European market. This, combined with the ending of many lease contracts, means charging point operators (CPOs) and policy makers expect a transition of the Dutch EV fleet from PHEV to FEV in the near future. This is supported by EV sales trends in the Netherlands which show that in the last year more than 90% of EV sales have been FEVs.

This leads to the question of whether the current public charging infrastructure is capable of accommodating the new composition of the EV fleet. To answer this question, a simulation model that incorporates the differences in charging behavior between small and large battery-sized vehicles is required [8,9], thereby representing PHEVs and FEVs, respectively.

Many simulation models exist today on the topic of EV [9–16]. However, to the best of our knowledge, those models are generally not validated or only validated using small amounts of data [9,17–19]. Furthermore, these models do not incorporate differences in charging behavior related to battery size.

Therefore, in this research we examine the effects of the transition of EV users from PHEV to FEV on charging behavior using real world data [20]. From this analysis, a behavior transition

equation is developed to transform any PHEV user type to an equivalent FEV user. Simulations with different FEV transitions were performed in an agent-based model (ABM) that included real-world charging data. From this, conclusions regarding the effects of charging infrastructure performance were drawn [6,21,22].

## 2. Literature Overview

Research on the influence of battery size on the charging behavior of EV users is still scarce. Zoepf et al. [23] conducted research concerning PHEVs with some aspects of fuel consumption versus battery use for varying battery sizes. They concluded that fast chargers are of little added value for users with a small battery.

Wei et al. [24] presented a tool to estimate fast-charging demand and sample results on a current and future EV scenario. Their results showed the interaction of battery size, frequency of charging, and energy needed per charging transaction.

While energy per charging transaction increased with battery size, the overall electricity demand per vehicle decreased with larger batteries. This was due to less charging transactions, with more kWh charged per transaction. A reason for this may be that that larger battery FEVs tend to reach their destinations more often, which leads to less transactions, while, if needed, the transaction size is larger due to the larger battery size.

They used long-distance data and provided a table with interaction between battery size and number of charge scenario results. They considered battery sizes of 80, 150, and 300 miles and showed that the demand in kWh from fast charging per vehicle decreased as the battery size increased. The results showed how battery size may interact with charging behavior, particularly the share that will be fast-charged versus regular charging; but does not shed light on the actual charging behavior on public (slow) chargers, which is the focus in this study.

Franke and Krems [8] performed a study that focused on user-battery interaction of EV users based on the concepts of how mobile phone users charge their phones. While this research provided results on how EV users cope with battery size, it did not provide insights usable in simulations.

Tal et al. [25] presented a survey of more than 3500 PHEV owners, conducted in California in May and June 2013. Their findings included the following. There was a low correlation between (i) the need for charging and (ii) actual charging transactions for low-battery PHEVs, mainly due to public charging availability, as reported by the drivers. PHEV drivers with higher battery capacity and FEV drivers charge more often and are more positive on charging opportunities in locations where low-battery PHEVs did not charge. They suggested that users with a low-battery PHEV may not have a high enough incentive to charge their car often.

Concluding, while interest is clearly being shown in the influence of battery sizes and car types on the behavior of EV users, little research is done in this area. Moreover, a real understanding of the effects of battery size on total charging infrastructure performance has not been found in literature so far.

## 3. Method

This research builds upon the simulation of electric vehicles activity (SEVA) model, a data-driven agent-based simulation model. A full description of the model can be found in [26]. In this model, the EV users are the agents and the charging points (CPs) are part of the environment. For each individual agent, rules for charging behavior of agents are generated based on the agent's historical data.

For each individual agent, the geospatial part of charging behavior is captured from the charging data by clustering nearby charging points, where the agent displays a similar activity pattern in time. For instance, evening charging of an agent may be done on several nearby charging points in the vicinity of the agents' destination. Each agent present in the model may have more than one cluster of CPs, and each cluster is a unique attribute for that specific agent.

Each cluster has a geospatial center based on the weighted average of the transactions at the CPs in a cluster, indicated by the X in Figure 1. The maximum distance of a used CP and the center of the cluster is regarded as the walking preparedness of the EV user, with a minimum of 150 m. Each CP within the radius of the walking preparedness measured from the center of the cluster is a potential candidate for charging of the EV user as it is assumed to be willing to walk from this CP to its destination at the center. EV users may either select a CP based on a preference scheme, derived from charging data distributions, or based on distance, which implies the nearest available CP from the center.

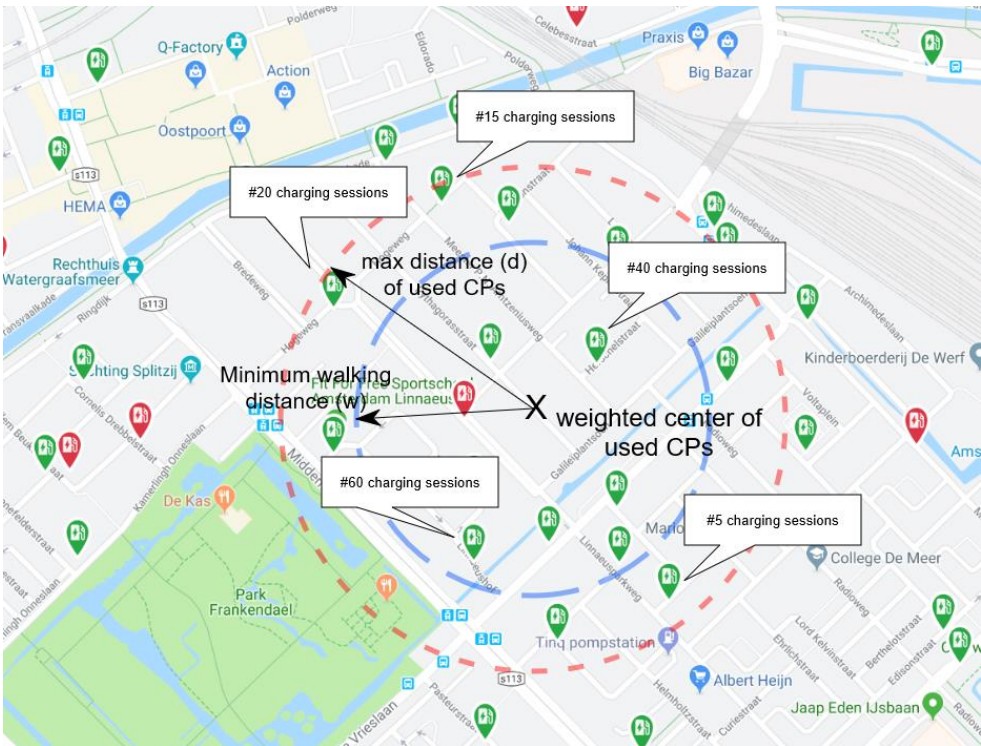

**Figure 1.** Illustration of a cluster of an agent. The cross indicates the weighted center, the boxes contain the number of sessions at the charging station, red dash indicates cluster region based on maximum distance between the center and charging point (CP), and the blue circle indicates the minimum walking distance in the simulation (picture adjusted from www.oplaadpalen.nl).

Based on the illustration in Figure 1, this distribution normalized is based on 60, 40, 20, 15, and 5 sessions at the different CPs. Only the five CPs which have been used are taken into consideration if an agent selects CP based on preference. For distance-based selection, all other CPs within the cluster (the areas within the red dashed circle) are taken into consideration. Note that the distributions are updated after successful connection, such that new CPs can be added to the preferences.

At initialization, for each individual agent the temporal behavior in the model is captured from the data in terms of distributions of connection and disconnection times. The connection time is the connection time at a CP, conditioned on the selected cluster. The disconnection time is the time between charging sessions, conditioned on the next cluster that is selected. As such, there is an assumed relation between the current and future cluster.

The charging behavior in the model is tuned and validated by comparing the simulation results for a training and test set of charging behavior. The validation metric compares the activity patterns of both charging point and EV-user activity. The metric is calculated by taking mean of all absolute differences of an activity pattern of an EV user of CP in the simulation and the corresponding pattern in the training or test data set, divided by the maximum differences. As such, the metric corresponds to the overall quality of the model. The metric also accommodates newly added charging points during

simulation that are not present in the training set. The metric applied on the simulation results showed that the charging behavior dynamics are well covered in the simulation. This research builds upon the validated charging behavior with addition of the transaction volume.

The dataset available for this research contains more than 2.7 million charging transactions on 3520 public level two AC charging points with two sockets, mostly 22 kW $2 \times 16$ amp three phase, of 66,160 users in the four largest cities (Amsterdam, Rotterdam, The Hague and Utrecht) in the Netherlands [27]. Table 1 provides an overview of the data in terms of users and sessions. The left side of the table contains the number of users in a year given a user type and the right side contains the same information for the number of transactions. In the table, a breakdown is given for the regular user type based on the total number of sessions per user, not per year. A detailed description of the data can be found in [7].

**Table 1.** Breakdown of sessions per year per user type.

| Year/User Type | Users | | | Transactions | | |
|---|---|---|---|---|---|---|
| | 2014 | 2015 | 2016 | 2014 | 2015 | 2016 |
| regular (0–10 sessions) | 8430 | 15,446 | 28,693 | 19,934 | 34,072 | 78,596 |
| regular (11–20 sessions) | 2161 | 3991 | 5375 | 14,161 | 26,022 | 54,886 |
| regular (21–50 sessions) | 2224 | 3996 | 5455 | 31,616 | 56,292 | 122,938 |
| regular (51–100 sessions) | 1239 | 2353 | 3209 | 44,076 | 71,084 | 163,106 |
| regular (101–250 sessions) | 1444 | 2944 | 3745 | 100,155 | 185,289 | 416,226 |
| regular (250–Inf sessions) | 1384 | 2296 | 2241 | 213,516 | 398,925 | 475,535 |
| Car2GO | 421 | 576 | 411 | 70,482 | 70,043 | 72,301 |
| taxi | 134 | 180 | 301 | 4182 | 18,389 | 29,979 |

Not all data was useful for model development. Research has shown that a large percentage of EV users in such a dataset is from irregular users that use the public charging infrastructure in a nonhabitual manner [28].

In order to abstract behavioral rules from the data, each EV user needs to have at least 20 transactions in a local area and at least 10 at a charging point (CP) in order to be part of the cluster. To assure generalization, special user types such as car-sharing cars and taxis are filtered out at the initialization of the model. In this paper, data from 1 January 2014 to 1 January 2016 was used as the training set and the data of 2016 was used as the test set. To validate the EV-users behavior after simulation they were requirement to be present in both the training and the test set. As a result, a selection of 2177 users were selected for model development.

To simulate the change in behavior due to a change in batteries, a clear understanding is needed as to how the connection and disconnection distributions change. It is also arguable that the clusters of an agent may change as their batteries change. A user might have fewer or more regular charging locations depending on its battery size. However, it also seems likely that a user would keep some, if not all, of its centers, as the user still drives the same routes and visits the same locations when it gets another car.

To decide exactly which aspects of the agents are crucial in capturing the change of battery size, a data analysis focused on the differences in batteries was performed. Next, based on behavioral properties on charging data, three types of EV were distilled from the data: (1) PHEV, (2) small-battery FEV (low FEV), and (3) large-battery FEV (high FEV). The differences in charging behavior were made explicit for modelling by drawing distributions on connection and disconnection to charging points and location-based behavior.

In the next step, a factor transform (FT) function was developed to apply the EV transition to behavior properties. Subsequently, the FT function was implemented in the ABM to simulate transformations of the current EV fleet from PHEVs and small-battery FEVs to large-battery FEVs. Finally, simulations were performed with different transition probabilities to reveal insight into different future scenarios. From the data derived from simulations, effects on typical key performance indicators

(KPIs) of charging infrastructure were analyzed [28]. This led to conclusions and recommendations for policy makers and CPOs.

## 4. Results

### 4.1. Battery Size Analysis

To identify differences in behavior between PHEV and FEV and the effect of battery size, a distinction between these types of EVs is required. A large amount of the EV users in the dataset can be classified as either owning a PHEV or a FEV with a classification method that relates maximum transaction size to EV car type [29].

The classifier considers the largest transaction volume and charging speed of an EV user and compares this with known properties of EV models to classify a user. In case of doubt, the user gets the label "unknown". These unknown EVs are filtered out in this analysis and the simulation model. For assessing the battery capacity of any given agent, we are assuming that every agent will approximate maximum charging and discharging during one of its charging sessions.

For PHEV this is more likely so, given that PHEVs have an Internal combustion engine (ICE) as backup. For FEV's this may lead to an underestimation of the battery capacity, also given that FEV's are known to maximize discharging to approximately 10% State of charging (SOC) in order to prevent battery degradation. As such, large-battery-size FEVs tend to show an underestimation of the battery size in our model.

Previous research on differences between PHEV and FEV that used this assumption has shown that the peaks in the histograms of kilowatt-hours (kWh) charged show comparable results with battery capacities of typical electric vehicles in the market [29]. This can also be seen in Figures 2 and 3. For instance, in Figure 3 a peak around 21 kWh may indicate a Nissan Leaf. In this Figure the underestimation of large-size-battery FEVs can also be seen. For instance, the Teslas with >75 kWh battery sizes do not peak as much as the lower-battery-size FEVs and PHEVs. Yet, the consequence of our assumption for this research is limited as we transform from transaction volumes of one type of battery to another, rather than transforming the battery size itself.

Different cars can be used with the same charging card ID, because a charge card is not bound to a car but to a user. For instance, an EV user may incidentally use its charging card to charge, for example, a hired car or EV from a colleague. Therefore, it was decided not to set the battery size of a user as the maximum kWh observed in all charging transactions, but as a percentile to filter out the top percentage of the transactions. A deeper analysis revealed that in the 98th percentile there was a good balance between filtering outliers while not filtering too many regular transactions.

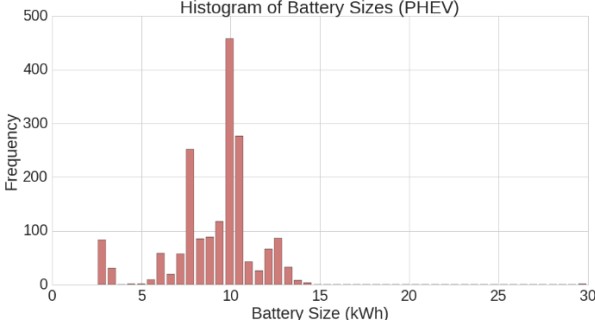

**Figure 2.** The spread of battery capacity for all (1727) plugin hybrid electric vehicles (PHEV) users present as valid agents in the simulation of electric vehicles activity (SEVA) model.

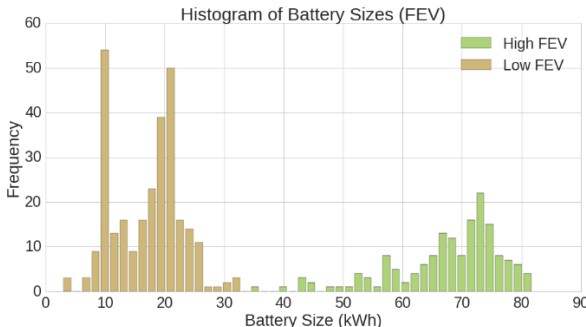

**Figure 3.** The spread of battery capacity for all (445) full electric vehicles (FEV) users present as valid agents in the SEVA model.

Based on the results of the classifier and the battery-size calculation, the spread of battery size of PHEV and FEV of (2172) users present in the simulation model can now be displayed (see Figure 2 (PHEV) and Figure 3 (FEV)). The FEV users are clearly split into two groups, one with low and one with high battery capacity, while the PHEV users are all close to a mean of 10 kWh. The outlier of 30 kWh may be a wrongly identified FEV (e.g., the BMW i3 with large battery).

While there are no models available with battery sizes between 33 kWh (BWM i3) and 70 kWh (Tesla Model S) in the period up to December 2016, we do see some occurrences of battery sizes between those values in Figure 3. This is caused by users that mostly charge their car before the battery is fully empty. Given that the actual battery size is unknown in the available dataset and that the behavioral properties of the EV users relate to the transaction volume rather than the battery size, we decided not to rescale the Figures 1 and 2 to known EV battery sizes.

Based the current results, the users were split in the dataset into three groups based on their car type and battery size. Namely PHEV (1727 users), low FEV (283 users) with low battery capacity (up to 33 kWh) and high FEV (162 users) with high battery capacity (over 33 kWh). The low FEV group includes models such as Nissan Leaf; the category over 33kWh includes Teslas.

### 4.2. Geospatial Charging Behavior for Different Battery Sizes

In this section, the differences that can be found between PHEVs, low FEVs, and high FEVs are described and tested on significance. First the differences in geospatial behavior are analyzed. This contains the differences in number of centers that this user type on average has, which can be related to the number of locations where EV users typically charge. Next, the clusters size, being the number of CPs of a center, is analyzed. Last, given that the cluster size relates to the distances of CPs in the cluster, the walking preparedness is also considered in this analysis. This is defined as the maximum distance between two CPs in a cluster.

In Figures 4–6, the mean and 95% confidence intervals of the number of centers, number of CPs per cluster, and walking preparedness are shown.

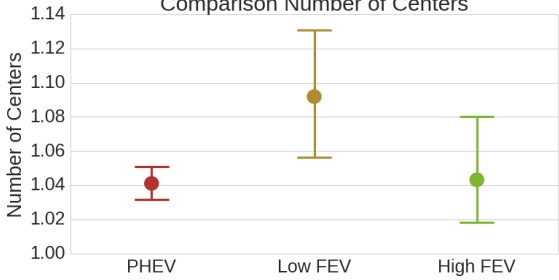

**Figure 4.** Comparison of number of centers between the three battery categories.

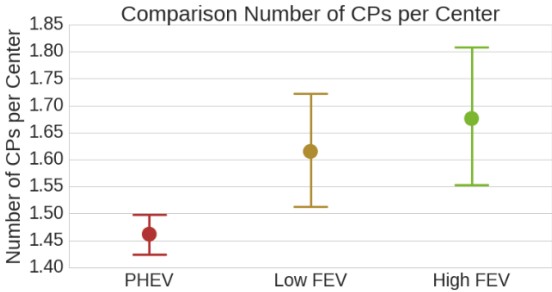

**Figure 5.** Comparison of number of charging points per center between the three battery categories.

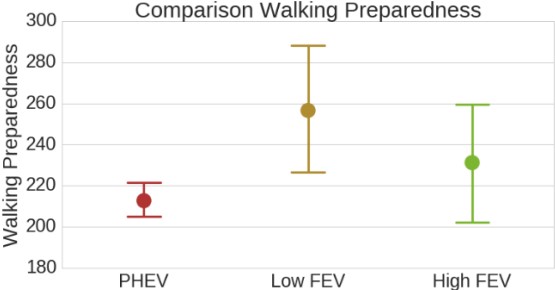

**Figure 6.** Comparison of walking preparedness between the three battery categories.

For each combination seen in these figures, we performed a two-sample independent *t*-test, assuming unequal variances between the two samples. The *p*-values resulting from these tests can be found in Table 2, where significant *p* values are emphasized in green.

**Table 2.** For the three geospatial variables in the first column, a two-sided *t*-test for each combination is carried out.

| Property | FEV (Low) & PHEV | FEV (High) & PHEV | FEV (Low) & FEV (High) |
|---|---|---|---|
| Number of centers | 0.001 | 0.902 | 0.085 |
| Number of CPs per center | 0.004 | 0.001 | 0.493 |
| Walking preparedness | 0.000 | 0.212 | 0.297 |

Possible explanations for the found differences are the following. As PHEVs have less incentive to charge often, they are not required to search alternative CPs when their preferred CP is occupied. This could be a reason that the walking preparedness and the number of CPs per cluster are lower for PHEVs. The low FEVs have more centers than the other two categories, possibly because in this category the need to charge is highest, thus they seek alternative charge locations.

From this we conclude that regarding geospatial behavior, most differences are present between low-battery FEVs and PHEVs. Given that the simulation is set up to transform the PHEVs to high-battery FEVs, the only factor to transform is the number of CPs per center.

### 4.3. Temporal Charging Behavior for Different Battery Sizes

The temporal charging behavior is based on the connection and disconnection distribution of an agent given per cluster of CPs. Before analysis, a cutoff on extremely short sessions less than 5 min and long connection durations of 5 days and disconnection durations of 40 days was performed to filter outliers and errors. Afterwards, the mean behavior per unit time was calculated by polynomial fit.

In Figure 7, a normalized frequency plot of the mean connection duration distributions (a) and the differences between the distributions (b) for each of the battery categories is displayed. The differences in normalized frequencies were specifically added to emphasize the differences between behavior in the distributions, since these differences are to be transformed during simulation.

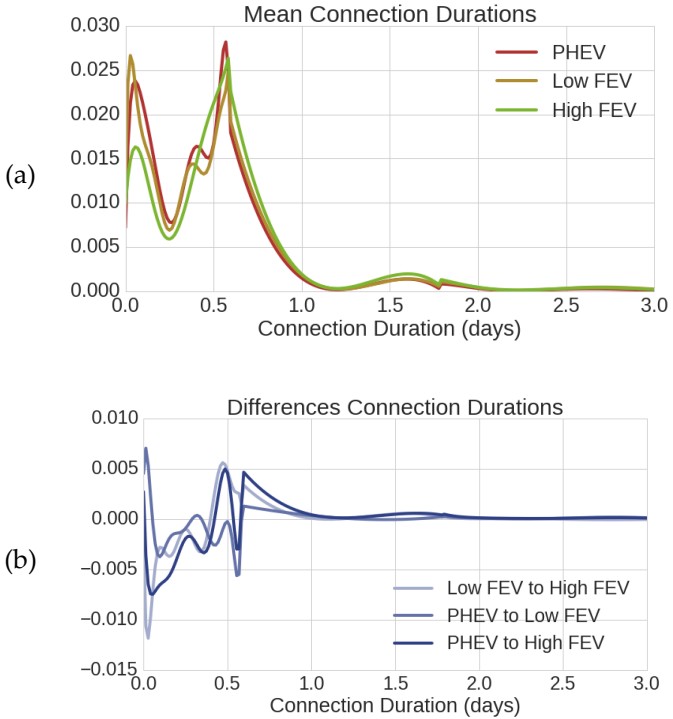

**Figure 7.** The mean connection duration distributions for the different battery groups, where (**a**) shows the normalized frequency plot of the fitted distributions and (**b**) the differences between those distributions.

From this, it is found that the high FEV users have a much lower first peak around 0.05 days (first hour), which suggests that high battery users are less likely to connect for (very) short durations. This may be explained by the idea that high-battery EVs have a longer driving range and as such do not need to stop and charge for a short time in between driving and may prefer to wait and charge for a more solid duration. It can also be observed that low FEV and PHEV categories have a peak at both 0.35 and 0.55. We hypothesize that the 0.35 peak is due to the users charging at work (roughly 8 h), while the 0.55 peak might point to users charging at home overnight (roughly 13 h), also found in [28,30]. This peak is absent in the high FEV group, which may indicate that large-battery FEVs may not specifically charge at work as much as PHEVs and low-battery FEVs.

The arrival time distributions indeed indicate a lower number of high FEV users that start their charging session at typical daytime or office hours (Figure 8).

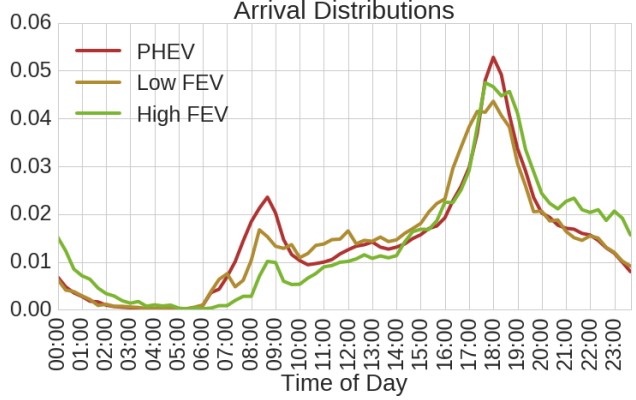

**Figure 8.** The arrival time distributions for the three different groups.

In Figure 9, the normalized frequency of the mean disconnection duration, being the time between two subsequent sessions, distributions is displayed. Note that the high FEVs shows longer

disconnection durations. High FEVs are found to have 49% of their disconnections longer than a day, while in both PHEVs and low FEVs this is only 30%. The figure shows peaks at 0.4, 1.4, and 2.4 in the high FEV pattern, indicating a disconnection of roughly nine hours plus zero to two days. This indicates that high FEV users tend to skip transactions and reconnect at the same time or later each day. This corresponds with the finding that high FEVs have a lower mean number of weekly sessions than the other categories.

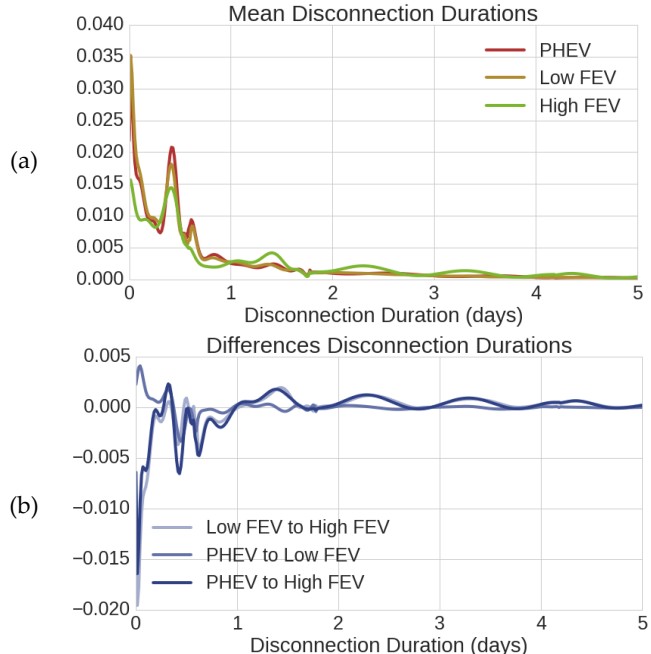

**Figure 9.** The mean disconnection duration distributions for the different battery groups, where (**a**) shows the normalized frequency of the fitted distributions and (**b**) the differences between those distributions.

For the lower battery categories, the highest peak is for very small disconnection durations of less than an hour. This may indicate that those users most often charge again right after reaching their destination. A deeper analysis of the differences between the distributions using the Hellinger distance showed that the high-battery FEV tend to differ more from the other two categories by disconnection than by connection duration. Low FEVs and PHEVs are very similar in both disconnection and connection patterns. Next, high FEVs differ from the other two groups by less charging transactions, which on average are longer and less frequent.

### 4.4. Transformation of EV User Behavior

Having a clear insight in the differences in charging behavior, a transformation function can be developed that enables transforming a user from one category to another. This can then be implemented in the simulation model to evaluate how this plays out in the utilization of charging infrastructure. The SEVA model used in this research simulates the behavior of an agent mainly by the use of the disconnection and connection duration distributions. The combination distributions of both results in arrival patterns and the number of weekly sessions. Therefore, the connection and disconnection distributions are the main subjects to the transformation.

However, in this research, we choose not to consider the differences in center characteristics with the transformation for a reason. Namely, the differences, although significant, are a lot smaller than other differences, and the manipulation of centers is not straightforward, since they are directly pulled from the dataset with the clustering of CPs.

For the factor transformation (FT) we use the fitted means for the connection and disconnection distributions, as seen in Figures 7 and 9. The transformation can be defined by the following equation:

$$w_i = \begin{cases} v_i \cdot \frac{t_i}{o_i} & \text{if } o_i > 0 \\ 0, & \text{otherwise } i = 0. \end{cases} \tag{1}$$

where $w_i$ is the new value bin $i$ will take, $v_i$ is its old value, and $t_i$ and $o_i$ are the values in bin $i$ for the target distribution and the origin distribution, respectively. The term $\frac{t_i}{o_i}$ can be seen as the transform factor of the $i$-th bin. For each user transformation from PHEV to high FEV, $t$ would represent the high FEV (dis)connection duration distribution and $o$ the PHEV distribution.

## 5. Simulation Setup

### 5.1. Extension of SEVA Model

To implement the transformation of an agent from one battery category to another, the SEVA model needs several upgrades. First, agents should now have an attribute containing information on whether the agent is a PHEV, low FEV, or high FEV. Second, a transform probability which defines for each agent the probability of transforming from PHEV to FEV is added as a parameter of the model.

To implement the transaction volume in the model for a session of an agent, the whole population of a single battery category is used as predictive information on transaction volume for an agent within this group.

The transaction volume is modelled as follows. For the whole category, a heatmap of probability of transactions given the connection duration is set up (see Figure 10). In this figure, the color scale represents the probability given the connection duration and charging volume. It can be seen that this probability peaks at the origin (0,0), which is reasonable since with zero connection time no charging takes place. During simulation, the model samples a charged amount belonging to the simulated connection duration according to the probabilities at this connection duration.

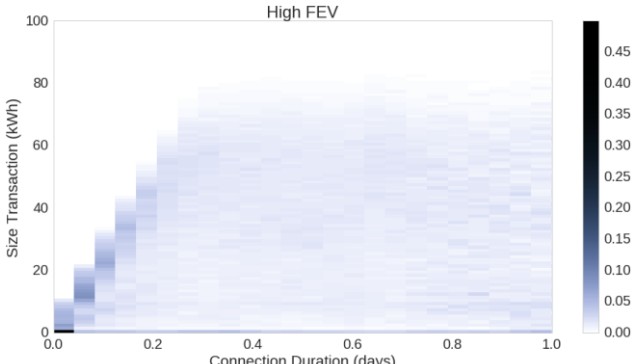

**Figure 10.** Heatmap of probability of a transaction volume given a connection duration. The values are probabilities. The color corresponds to probability values.

### 5.2. Simulation Metrics

The purpose of this study is to observe the change in demand on the charging infrastructure given the transition of battery size. Previous research revealed important key performance indicators on charging infrastructure for various stakeholders [28,31]. The following indicators were chosen to analyze from the simulation results: (1) Average connection duration per CP per week; (2) Average number of unique users per CP per week; (3) Average number of charging transactions per CP per week; (4) Average kWh charged per CP per week.

*5.3. Simulation Procedure*

The simulation procedure is set up to research the effects of the demand on the charging infrastructure as a transition takes place from a population fully consisting of PHEVs to one fully consisting of large-battery FEVs. As such, the simulation contains the 1727 PHEV agents and is performed in five simulation runs of one year with those agents, keeping track of the system measures. For each simulation run, every agent has a probability to be transformed to a high-battery FEV at the start of the simulation. This percentage varies from 0% to 100%, with a step size of 20%.

## 6. Simulation Results

The values of the system measures in the case study are plotted against the probability to transform in Figures 11–14. The error bars indicate the variance over all CPs and all runs.

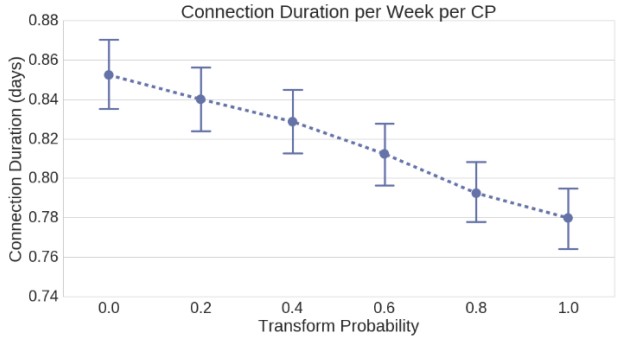

**Figure 11.** Weekly connection duration related to transform probability.

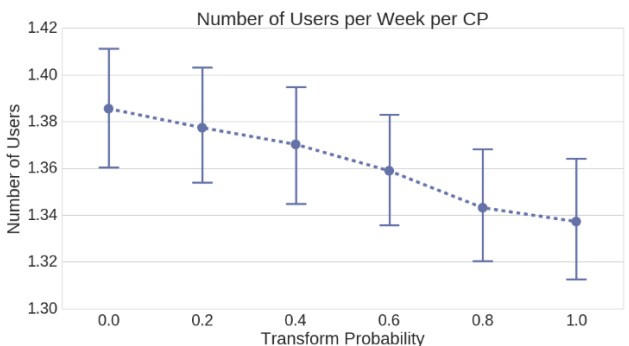

**Figure 12.** Weekly number of users related to transform probability.

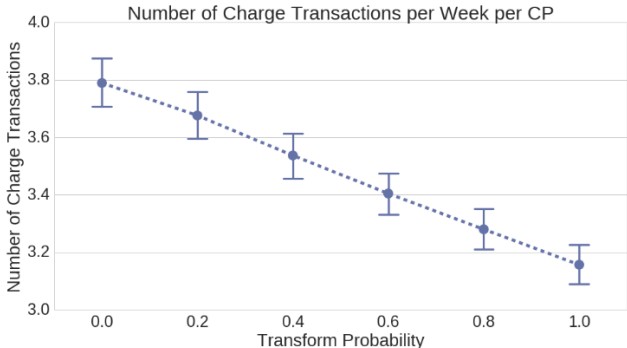

**Figure 13.** Weekly number of transactions related to transform probability.

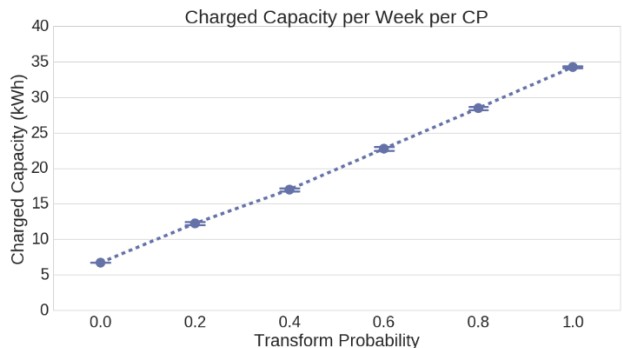

**Figure 14.** Weekly transaction volume related to transform probability.

There is a significant decrease (17%) in the number of charging transactions per week and the connection duration per week, which is as expected. This also confirms that the FT of disconnection and connection distributions does capture the difference in the number of charge transactions per week. The number of users per week also decreases, but not significantly over the scope of this transformation (0.7%). This decrease is not as strong as expected, which can be explained by the fact that the model does not decrease the number of CPs to choose from with the transformation. Each CP still has the same chance of being selected as it had before the transformation. The increase shown could be due to an agent having to deviate from its first choice less often, since CPs are less often occupied. Lastly, the total energy uptake and the average per CP per week on charged kWh increased significantly (80% and 70%, respectively) for every step. The reason for this may be found in the idea that PHEVs tend to use their full battery each day, and their transactions are limited by battery size rather than daily trip size, whereas FEVs do not have this limitation. The increase in total energy uptake by EVs at the full transition (probability 1.0) to large-battery FEV is ~550%.

## 7. Conclusions

This study presented a simulation model for the transition of EV user charging behavior. The model is an extension of the existing SEVA model (currently under peer review). The transformation of EV user charging behavior due to increase of battery size was performed based on data analysis of actual EV users' transactions.

From this case study, we see the utility of the CPs, and thus of the charging infrastructure, increases as the battery size of the population increases. The connection times per CP decreases, while the kWh charged at those same poles increases. This would indicate that, as a transition to higher batteries takes place, first the efficiency of charging infrastructure increases, and second, less charging infrastructure would be needed to facilitate the EV population.

The number of unique users per CP and the decrease in connection times would also be positive for EV users, as this implies that the CPs are available more often. Yet, the current transformation function could be improved on the CP selection process, which may affect this metric as well.

There are some drawbacks as well. As we have seen, high FEVs charge more at night times, and this could cause a higher peak on the energy demand as a higher fraction of the population starts charging at night. However, this peak might be shifted towards a later time using smart charging, as generally not the entire night is needed for a full recharge. Implementation of smart charging in future research and extension of this model would therefore be a logical next step.

Overall, the results of the case study indicate a decrease in demand on the charge infrastructure as battery sizes increase and the number of EVs stays the same, which is beneficial for most involved stakeholders.

**Author Contributions:** J.R.H., M.L. initialized and designed this research, I.V. conducted the research and analysis, I.V. wrote the code, I.V. wrote the thesis, J.R.H. and M.L. supervised the research, J.R.H. wrote the paper, M.L. and R.v.d.H. edited the paper.

**Funding:** This work is part of the doctoral grant for teachers with project number 023.009.011, which is (partly) financed by the Netherlands Organization for Scientific Research (NWO). The data is provided by the G4 cities of the Netherlands. Finally, this research is part of the IDOLAAD project funded by Stitching Innovatie Alliantie (SIA).

**Conflicts of Interest:** The authors declare no conflict of interest.

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
