# Peer review of "Simulation of Future Electric Vehicle Charging Behavior—Effects of Transition from PHEV to FEV"

_wevj, doi:10.3390/wevj10020042_

Round 1
Reviewer 1 Report
This report shows simulation results of EV charging behavior using Dutch charging point data for the future charging infrastructure. It was an important study using complex real data.
However, the results were not confirmed because agent-based simulation and geospatial analysis were not well described in the report.
page 2, line 82. Since the explanation about the simulator (EVSA) was omitted, it was not possible to confirm the main results of this paper. It is necessary to explain the process leading to the derivation of the results.
page 4, line 153. The explanation about geospatial analysis is not enough. In order to understand Figure 3-5, the author needs to explain the positional relationship between cluster, center, CP, and "Preparing for walking".
The validity of the results (Figures 10 to 13) could not be confirmed because the structure of the simulator (EVSA) and the input data (Figures 3 to 5) are not well explained.
Other comments;
Please enter the number of data used in this analysis. Breakdown of 5.6 million data (page 2, line 85).
“user will at least once charge from 0% to 100% SoC” (page 3, line 123) is an unrealistic assumption. Does this assumption affect the outcome?
Does the CP include both private and public chargers?
The color scale in Figure 9 has no description.
Is the total number of PHEVs and FEVs constant? If so, is energy consumption also constant in the simulation?
Author Response
Dear Reviewers,
First of all we would like to thank you for the evaluation of our manuscript by all reviewers. The comments made are both clear and reasonable which helped us improving the quality of the paper.
In the attached document , you find a detailed response to each remark made.
Sincerely the Authors.

Reviewer 2 Report
This paper investigates the effects of the transition from PHEV to FEV with higher battery capacity. I think this paper has some interesting ideas, but also have some drawbacks:
1- The terms "connection and disconnections distributions" need to be defined properly.
2- There is an assumption in line 124, which is "charge from 0% to 100% SOC". I think this assumption is not realistic. As far as I know, the range is usually between 20% to 80% SOC. Please clarify that.
3- The definition of "walking preparedness" is not clear.
4- In Figure 6 (a&b), what is the y-axis represent? and why there are negative values?
5- What are the chargers types and charging rates investigated in this study?
6- Also, there are some blank spaces between words such as in line 42 and 44.
Author Response
Dear Reviewer,
First of all we would like to thank you for the evaluation of our manuscript by all reviewers. The comments made are both clear and reasonable which helped us improving the quality of the paper.
In the page below, you find a detailed response to each remark made.
Sincerely the Authors.

Round 2
Reviewer 1 Report
I was able to image the simulation by adding a description of the model. However, the diagram you described in reference [26] is more helpful. It is better to add some figures to Chapter 3 for the reader's understanding.
Author Response
Dear Reviewer,
Indeed we understand this concern. We added a new illustration based on a picture in the original Arxiv paper, but now with Google maps underneath and real CPs. We also improved the text by including the elements from the added picture in the text. With this new illustration added we indeed believe the description of the simulation model is clear for the reader.
We thank you reviewer for keeping us as sharp as authors.
Best Regards
Reviewer 2 Report
I do not have any farther questions.
Author Response
We thank the reviewer for the acceptance of the paper.
Best regards the authors